# Determining the Potential of *Haematococcus pluvialis* Oleoresin as a Rich Source of Antioxidants

**DOI:** 10.3390/molecules24224073

**Published:** 2019-11-11

**Authors:** Mari Carmen Ruiz-Domínguez, Carolina Espinosa, Adrián Paredes, Jenifer Palma, Carolina Jaime, Carlos Vílchez, Pedro Cerezal

**Affiliations:** 1Laboratorio de Microencapsulación de Compuestos Bioactivos (LAMICBA, acronym in Spanish), Departamento de Ciencias de los Alimentos y Nutrición, Facultad de Ciencias de la Salud, Universidad de Antofagasta 02800, Antofagasta 1240000, Chile; carolina.espinosa@uantof.cl (C.E.); jenifer.palma@uantof.cl (J.P.); pedro.cerezal@uantof.cl (P.C.); 2Laboratorio de Química Biológicas, Instituto Antofagasta (IA, acronym in Spanish), Universidad de Antofagasta 02800, Antofagasta 1240000, Chile; adrian.paredes@uantof.cl; 3Atacama Bio Natural Products S.A., Vía 5 Esq. Vía 9, Bajo Molle, Iquique 1100000, Chile; carolina.jaime@gmail.com; 4Algal Biotechnology Group, CIDERTA-RENSMA and Faculty of Sciences, University of Huelva, 21007 Huelva, Spain; cvilchez@uhu.es

**Keywords:** *Haematococcus pluvialis*, oleoresin, astaxanthin, antioxidant capacity, supercritical fluid extraction, phenol content, viscosity

## Abstract

*Haematococcus pluvialis* is known to be a natural source of antioxidants for numerous applications. In this study, an oleoresin rich in carotenoids extracted by supercritical CO_2_ treatment of *H. pluvialis* was extensively characterized for its antioxidant capacity. Carotenoid content, fatty acid profile, total phenol content, antioxidant capacity, and viscosity of the oleoresin were determined with the aim of ascertaining the potential of the oleoresin in terms of its antioxidant content for food applications. The oleoresin contained 96.22 mg/g of total astaxanthin (which includes free astaxanthin and astaxanthin esters) and mostly included unsaturated fatty acids (~78% of total fatty acids). High total phenol content and ferric reducing antioxidant potential indicated high antioxidant capacity, but oxygen radical absorbance capacity was lower compared to the oleoresin samples obtained from other species. The oleoresin was a non-Newtonian fluid since it had shear-thinning (pseudoplastic) and shear-thickening (dilatant) flow. Therefore, the *H. pluvialis* oleoresin is a potential alternative in developing functional ingredients for designing healthy food products. To the best of our knowledge, this is the first study that has reported an extensive characterization of the antioxidant properties of a microalgal oleoresin obtained by means of supercritical CO_2_ fluid extraction.

## 1. Introduction

Recent consumer trends pertaining to consuming natural foods for improving health are changing because of a newfound interest in the origin and manipulation of food products. Hence, the food industry is developing new food products enriched with functional ingredients that may provide health benefits beyond the traditional nutrients it contains [1]. Microalgae are an important source of such functional ingredients with several potential biotechnological applications [2]. Microalgae are a large group of photosynthetic organisms with an efficient unicellular structure, which allows them to convert solar energy into chemical energy. They have been extensively studied and found to contain a broad range of high-value bioactive compounds with important applications in food, cosmetic, pharmaceutical, and biofuel industries [3,4], and these bioactive compounds have been recognized for preventing a variety of diseases and maintaining good health in humans [5,6]. Carotenoids, in particular, are the most extensive class of pigments synthesized by microalgae. About 30 of them play a direct role in light-harvesting and energy transfer during photosynthesis [7], while others are involved in the defense mechanism against radical oxidative species (ROS) that cause photo-oxidative damage [8]. Among the variety of species of microalgae, *Haematococcus pluvialis* is widely known to be the best in producing astaxanthin, reaching up 3–5% of natural pigment by dry weight [9]. The special characteristic about this rosy, water-insoluble carotenoid is that it is a byproduct of β-carotene under stressful conditions during carotenoid biosynthesis [10] and is a much more powerful antioxidant than vitamins C and E (α-tocopherol) or other carotenoids such as β-carotene, lycopene, lutein, and zeaxanthin [11,12]. However, to the best our knowledge, no extensive studies have been published regarding the specific antioxidant and rheological properties of *H. pluvialis* oleoresins. The oleoresin samples analyzed in this work were obtained from *H. pluvialis* grown in northern Chile, a place where solar radiation is one of the highest in the world. In this area, the daily solar radiation energy ranges from 9.5 kWh/m^2^ during summer to 4.5 kWh/m^2^ in winter [13], which are outstanding stressful conditions for triggering the synthesis of antioxidants in outdoor-growing microalgal cells. 

Astaxanthin has been known to have anti-inflammatory [14] and immune-enhancing properties that are beneficial for both humans and animals alike; it can also aid in preventing cardiovascular diseases such as arteriosclerosis [15]. In addition, it is widely used in fish feeds, particularly those for salmon, crustaceans, and even birds. Astaxanthin is well known for its red-orange color [4,16,17]. For the optimal extraction of carotenoids, many studies have used several methods that are different from traditional extraction methods [18,19,20]. In general, the most common green methodologies along with conventional extraction methods (i.e., atmospheric liquid extraction (Soxhlet extraction method) or maceration) are as follows: (i) microwave-assisted extraction, (ii) ultrasound, (iii) high-pressure homogenization (HPH), (iv) pressurized liquid extraction (PLE), (v) enzyme-assisted extraction, and (vi) supercritical fluid extraction (SFE), which is often based on the use of supercritical carbon dioxide (SC-CO_2_). The application of supercritical fluid extraction (SFE) for the recovery of valuable compounds is more attractive than other methods even after considering environmental protection [21]. Many bioactive natural products are thermally labile and can degrade during the use of traditional extraction methods. The use of SC-CO_2_ has been demonstrated to be effective for the extraction of bioactive compounds [22,23], thus preserving these biomolecules from degradation during the extraction process. In our case, the study material used is an oleoresin sample enriched in carotenoids (~10%wt) obtained from the microalga *H. pluvialis* by SC-CO_2_ treatment. Specifically, the use of carbon dioxide (CO_2_) as the main solvent for the extraction process is the most distinguishing factor, as SC-CO_2_ has a low critical point (T 30.9 °C and P 73.9 bar) and is a cheap, nonflammable solvent that offers an extensive tunable selectivity [21]. Furthermore, it is generally recognized as safe (GRAS) by the Food and Drug Administration (FDA) [21,24]. 

On the other hand, bioactive compound extraction from any biomass involves an increase in the susceptibility to degradation of these molecules (isomerization and/or oxidation reaction) during its physical processing and storage [11]. Therefore, oleoresin is produced with the aim of protecting such compounds from degradation by mixing bioactive compound extracts with different vegetable oils [25,26,27]. In addition, using encapsulation methods with specific matrices more efficiently improves the stability of high-value bioactive compounds for a long time [28,29,30]. To the best of our knowledge, this study is the first to report a broad-level characterization of an oleoresin enriched in carotenoids extracted from *H. pluvialis* by the SC-CO_2_ method and stabilized by using a mixture of oils. The objective of this study is to provide additional information about the antioxidant capacity of the oleoresin prepared from *H. pluvialis* and assessing its value for functional food applications. Therefore, we performed total and individual carotenoid analyses and determined the fatty acid profile, total phenol content, antioxidant capacity, and viscosity. We used the data to evaluate the potential application of the carotenoids present in these oleoresin samples as a functional ingredient and/or stable bioactive additive in food products. 

## 2. Results and Discussion

### 2.1. Carotenoid Content in Oleoresin Samples of Haematococcus Pluvialis

Table 1 shows the carotenoid content in oleoresin samples from the microalgal species *H. pluvialis* determined by the SC-CO_2_ method. This microalgal species has been widely known to be a natural source of astaxanthin, which accumulates in response to shifts in a variety of environmental factors [31]. In this study, of all the carotenoids, total astaxanthin (considered to be the sum of free astaxanthin and astaxanthin esters) was found to be of the highest concentration (astaxanthin esters: 96.22 mg/g of oleoresin; free astaxanthin: 6.82 mg/g of oleoresin) followed by β-carotene, canthaxanthin, and lutein (2.35, 1.68, and 1.12 mg/g of oleoresin, respectively). 

The total carotenoid content in oleoresin samples determined by Montero et al. [32] using n-hexane was 11.49%wt. Many studies have indicated that several carotenoids have a positive effect on human health including the ability to reduce damage to DNA, proteins, and membrane lipids and prevent the degeneration of the macula in adults [33,34,35]. *H. pluvialis* is still considered as a potential organism for the production of astaxanthin [31,36,37]. The microalgae is also reported to contain most of the carotenoids similar to astaxanthin in different forms (free, mono-, and diesters), which are astaxanthin esters in the predominant form (90–95% of all astaxanthin forms) [38]. It confirms our results obtained by analyzing *H. pluvialis* oleoresin. Moreover, this microalga contains other carotenoids, such as canthaxanthin and lutein [39,40], which were present in the studied samples together with β-carotene. Astaxanthin and canthaxanthin are recognized as more potent antioxidants and scavengers of free radicals than β-carotene [41,42]. In contrast to other oleoresins [30], the carotenoid profile reported in the present study is broader.

Several reports have dealt with the extraction improvement of astaxanthin from *H. pluvialis* by green extraction methodologies. For example, Reyes, et al. [12] focused on assessing and validating the use of CO_2_-expanded ethanol (CXE) for extraction of astaxanthin from *H. pluvialis* versus supercritical carbon dioxide (SC-CO_2_). They achieved a higher astaxanthin content of 62.57 mg/g under CXE conditions (7 MPa, 50 °C, and 50% ethanol (w/w)) than SC-CO_2_ extraction (53.48 mg astaxanthin/g under extraction conditions of 20 MPa, 55 °C, and 13% (w/w)). Molino et al. [43] reported through accelerated solvent extraction (ASE) a complete study to improve the astaxanthin recovery from *H. pluvialis*. They used a mechanical pretreatment (ball mill MM400^®^, Retsch, Germany) from the microalgal biomass to assess the effect in the astaxanthin recollected after ASE extraction. The maximum recoveries of 86% and 67% were reached at 100 bar and at 40 and 67 °C for acetone and ethanol. Thus, our study shows an oleoresin rich in astaxanthin even with values higher than those described in literature. 

### 2.2. Antioxidant Capacity and Total Phenol Content

The antioxidant capacity was determined by performing ferric reducing antioxidant potential (FRAP) and oxygen radical absorbance capacity (ORAC) analyses. The results of these analyses and the total phenolic contents are summarized in Table 2. 

Phenolic compounds are considered one of the most important classes of natural antioxidants and have received increasing attention from consumers and food producers alike for their health benefits [44]. The oleoresin in the present study contained 74.08 mg of gallic acid equivalent (GAE)/g of oleoresin extract (OE), which was slightly higher than other direct results reported for the microalga. A few studies have reported the antioxidant capacity of *H. pluvialis* oleoresin, but these studies used microalgal extracts. For example, Batista, et al. [45] reported total phenolic content from four microalgal extracts. Their values were found to be lesser than ours, with 9.2 mg of GAE/g of biomass from *Tetraselmis suecica* or 19 mg of GAE/g of biomass from *Arthrospira platensis*. Oleoresins from other microalgal species, such as *Scenedesmus obliquus* and *Phaeodactylum tricornutum* with less phenolic content than the oleoresin studied (between 59.25 and 42.16 mg GAE/g), have also been found to have potential applications in the food industry [46,47]. Moreover, another study that used *H. pluvialis* extracts obtained through the “green phase” and the “red phase” reported 1.89 and 0.54 mg GAE/g of biomass, respectively [48]. These values are lower compared to those reported in the present study. Compared to the other methods of determining antioxidant capacity, FRAP is the most commonly used [49,50]. 

We, following this method, determined the antioxidant capacity of our oleoresin sample as approximately 313.76 mg Trolox equivalent (TE)/g of OE. The values reported in our study are higher than those reported by Goiris, Muylaert, Fraeye, Foubert, De Brabanter and De Cooman [48]. They reported *H. pluvialis* values to be about 4.0 to 10.50 mg TE/g of biomass and for *Chlorella* sp., *Scenedesmus* sp., or *Phaeodactylum tricornutum* in the range of about 1.5 to 22.50 mg TE/g of biomass. On the other hand, the ORAC method produced the lowest value of 5.22 µmol TE/100 g of OE in our study. The ORAC values reported in other studies, functional food samples enriched with astaxanthin from other *H. pluvialis* strains, were higher than the values reported herein [51,52]. Other species such as *Arthrospira maxima* have also been considered important sources of high-quality nutrients, exhibiting higher ORAC values ranging between 90–519 TE/g of biomass [53]. 

Although the ORAC values were very low, the total phenolic content and FRAP results indicate *H. pluvialis* oleoresin to be an important functional ingredient of food. The use of different antioxidant assays is necessary to evaluate the potential use of an oleoresin sample. Previous studies have shown that the antioxidant ability of oleoresin is due of its electron transfer capability, which is determined by FRAP. However, the values determined by ORAC, DPPH, and ABTS assays, designed to measure the ability to transfer hydrogen atoms to neutralize a free radical, were low compared to those reported in other studies that used natural extracts [53,54,55,56]. This mechanism of action can be explained by the presence of astaxanthin in oleoresin, which is responsible for electron transfer. As at equilibrium, the enolic form of the ketone results in an ortho-dihydroxy-conjugated polyene system capable of acting as a chain break in the free radical reaction similar to the hydroxyl group of α-tocopherol [57,58]. As of today, only a few studies have reported the antioxidant capacity of microalgal oleoresins produced by the SC-CO_2_ method. The phenolic compounds identified while determining the total phenolic content are common, and all of them exhibited antioxidant activities [48,59]. Several studies have evaluated different solvents to increase the extraction of these molecules from microalgal biomass [48,56]. Microalgal strains such as *Crypthecodinium cohnii* (dinoflagellate) and *Schizochytrium* sp. (thraustochytrid) have been considered important microalgal sources of docosahexaenoic acid (DHA), and their antioxidant capacity has been reported to be approximately 140 mg GAE/g of biomass with *n*-butanol treatment [56].

### 2.3. Fatty Acid Profile of Oleoresin Samples from H. pluvialis

The fatty acid profile (FAMEs and fatty acid methyl esters) was determined by the acid esterification method, and the results were expressed as the percentage of fatty acid methyl esters with respect to total fatty acids in oleoresin samples (Figure 1). The oil present in the oleoresin samples originated from the microalga itself (*H. pluvialis* ~100%wt). Therefore, the total integrated FAMEs had the highest concentration of unsaturated fatty acids (~78%) as determined by GC-FID. These unsaturated fatty acids included eicosenoic acid (C20:1, 25.49%), linoleic acid (C18:2, 22.01%), elaidic acid (C18:1 *trans*, 15.01%), and oleic acid (C18:1, 12.52%). Eicosapentaenoic acid (C20:5n3, 1.28%) and arachidonic acid (C20:4n6, 0.43%) were also present in the oleoresin fatty acid profile. On the other hand, the saturated fatty acid palmitic acid (C16:0, 16.96%) was found in the maximum concentration. All these values provide valuable insight into the antioxidant profile of our oleoresin, thus emphasizing its health benefits. Other studies have also reported the fatty acid profile from the oil extracts of *H. pluvialis*, which is rich in polyunsaturated acids [60,61]. The majority of the fatty acids identified under control conditions were caproic (18%), oleic (16%), and α-linolenic acids (21%). α-Linolenic acid (C18:3n3) has been particularly found as a major fatty acid in *H. pluvialis* extracts [60,61] despite it remaining probably undetected in the oleoresin samples examined in the present study. Bustamante, Masson, Velasco, del Valle and Robert [30] reported a similar fatty acid profile in *H. pluvialis* using the supercritical fluid extraction (SFE) method, wherein the most abundant fatty acids were polyunsaturated (PUFA, 45.3%), followed by monounsaturated (MUFA, 34.6%) and saturated fatty acids (SFA, 20.1%). Other edible oils were also investigated for oleoresin production, and diverse fatty acid profiles with antioxidant properties were reported [25,30]. However, the reports on the oleoresin production using oil from the same microalga are rare. A fatty acid profile containing omega-3 (ω-3) and omega-6 (ω-6) fatty acids has been proven to have health benefits, such as mitigating hypertension and anti-inflammatory effects as well as slowing down macular degeneration, rheumatoid arthritis, and osteoporosis [62,63,64]. Hence, an oleoresin rich in PUFA and MUFA could further increase its positive effects when used as a functional ingredient in foods. 

### 2.4. Viscosity Behavior

Viscosity is a measure of intermolecular friction in a fluid and of its resistance to flow. In general, commercial oils behave as ideal Newtonian liquids, and their viscosity decreases exponentially with increasing temperature [65]. The main factors that affect emulsion viscosity include the degree of emulsion dispersion (usually fine-dispersion emulsions exhibit higher viscosity than coarse-dispersion emulsions, and the presence of air bubbles in the emulsion increases its viscosity), the viscosity of individual phases of emulsion, and emulsion lifetime [66,67]. The high viscosity of the outer phase has a positive effect on emulsion stability. The increasing oil content in the emulsion (from 10% to 22%) contributed to the increasing viscosity of emulsions and intensification of their pseudoplastic character [68]. Figure 2 shows the relationship between the shear stress (SS) and shear rate (SR) of the oleoresin samples at different temperatures. Figure 2A depicts SS and SR in the low-temperature zone (5 to 20 °C). The SR values obtained were 6.8, 10.2, 17.0, 20.4, and 34.0 s^−1^, while the SS values ranged from 22.06 to 59.29, 16.72 to 65.03, 8.12 to 49.86, and 6.48 to 34.75 Pa at 5, 10, 15, and 20 °C, respectively. Figure 2B shows the values obtained in the high-temperature zone (30 to 60 °C). The SR values obtained were 18.6, 27.9, 46.5, 55.8, and 93.0 s^−1^, while the SS values ranged from 1.45 to 18.40, 0.24 to 8.58, 1.76 to 4.88, and 0.11 to 2.14 Pa at 30, 40, 50, and 60 °C, respectively. According to the recommendations of the viscometer manufacturer, the measurements with torque values <10% were not included. These were SS values for T = 5 °C with an SR value of 34.0 s^−1^, and SS values for T = 50 and 60 °C with SR values of 18.6 and 27.9 s^−1^, respectively (Figure 2A,B). In both Figure 2A,B, with an increase in the SR values for each temperature, the SS values were also found to increase, which led to the apparent viscosity values decreasing with increasing temperature (Table 3). 

Nevertheless, at 5 and 10 °C, the oleoresin containing other carotenoids could be categorized as a shear-thinning fluid (pseudoplastic) since its flow behavior index (n) had a value of <1 [69,70] (Table 3). However, at temperatures ≥15 °C, the samples showed a shear-thickening flow (dilatant) since the value of n was >1.0. This is atypical behavior for this type of fluid, and there are very few studies that have reported this phenomenon. Roenningsen et al. [71] indicated that some types of oils behave differently, probably because they are highly biodegradable with very low total wax content and certain atypical wax compositions with high proportions of isoalkanes and cyclic alkanes. 

Furthermore, the data in Table 3 indicate that the viscosity and consistency index values decreased as the temperature increased. Similar behavior was reported by Atacama BioNatural Products Inc. (https://www.atacamabionatural.com/web/index.php) in its “Red Meal” product. It is an emulsion (astaxanthin content < 1.5%) intended for animal feed, where the values for viscosity and Flow behavior index decreased significantly from 8.89 ± 0.10 to 1.07 ± 0.72 Pa·s and from 0.99 to 0.15, respectively, indicating a pseudoplastic behavior for the same temperature range of the present study (5–70 °C). In another study conducted by Coupland and McClements [65] for commercial oil, the viscosity decreased exponentially when the temperature increased. Different results were obtained regarding some emulsions that did not fit into any of the categories. Some dispersions or emulsions exhibit various non-Newtonian behaviors depending on the SR. For example, the dispersions of latex particles exhibit Newtonian behaviors at a very low SR. If SR increases further, viscosity decreases. In this condition, the fluid exhibits a shear-thinning behavior (pseudoplastic). At a still higher SR, the suspension behaves as a Newtonian fluid. In this case, if SR increases further, the suspension viscosity increases. In this condition, the fluid has a shear-thickening behavior (dilatant) [72]. Belyamani et al. [73] described the rheological properties of oleoresins produced by mature trees of four southern pines native to North America (loblolly, slash, longleaf, and shortleaf). Their results indicated that these oleoresins were structured fluids that exhibited viscoelastic behaviors but differed in flow behavior. Slash pine oleoresin exhibited Newtonian flow behavior, while the oleoresin from longleaf and shortleaf pines showed pseudoplastic behavior, and the loblolly pine oleoresin showed Bingham fluid behavior with a yield stress of about 1.980 Pa. Temperature has a significant effect on the rheological characteristics of the fluid, concentrated food products, and oleoresins. The effect of temperature on apparent viscosity at constant SR can be described by the Arrhenius equation [73,74,75]. Figure 3 shows that the Arrhenius model gave a good description of the temperature effect on the apparent viscosity of oleoresin. For low temperatures (5–20 °C, Figure 3A), the constant behavior of each SR was 6.8, 10.2, 17.0, 20.4, and 34.0 s^−1^, and the corresponding determination coefficients (R^2^) were 0.946, 0.957, 0.947, 0.974, and 0.984, respectively (Table 4). 

Moreover, in the high-temperature range (30–60 °C, Figure 3B), the constant behavior of the samples at each SR was 46.5, 55.8, and 93.0 s^−1^, and the corresponding determination coefficients (R_2_) were 0.863, 0.934, and 0.992, respectively (Table 5). Table 4 and Table 5 show the activation energy (Ea) of the oleoresin samples ranged from 43.65 to 110.68 kJ/mol. These values are high compared to those of juices, pulps, and concentrate juices. For instance, for mango juice, the Ea ranged from 3.8 to 13.7 kJ/mol [76]; for whole blackberry pulp, Ea = 18.27 kJ/mol [74]; and for tamarind juice concentrate, Ea = 35.09 kJ/mol [75]. Nevertheless, these values of Ea for oleoresin were much lower compared to the oleoresins extracted from mature trees from four southern pines native to North America: slash pine, 706.39–766.60 kJ/mol; shortleaf pine, 586.31–662.36 kJ/mol; loblolly pine, 869.99–940.50; and longleaf pine, 667.41–712.82 kJ/mol [73].

## 3. Materials and Methods

### 3.1. Materials

The study materials consisted of oleoresin samples of natural carotenoid complex (~10%wt) extracted from the microalga *H. pluvialis* (Flotow) strain Steptoe (Nevada, USA) with the help of the SC-CO_2_ method. The oil obtained from *H. pluvialis* was used in the preparation of the oleoresin. Chilean Company Atacama Bio Natural Products Inc. (Iquique, Chile) kindly donated oleoresin samples to us. General details of the *H. pluvialis* production are offered on the company’s website (https://www.atacamabionatural.com/web/index.php/about-us/our-process). All chemicals used in this study were of analytical grade, except for those used for HPLC and GC analyses (high-performance liquid chromatography and gas chromatographic, respectively), which were of chromatographic grade (methanol, acetonitrile, ethyl acetate, water, and *n*-hexane). The carotenoids astaxanthin, lutein, canthaxanthin, and β-carotene, chemicals for the analysis of total phenols, cholesterol esterase, and reagents for determining antioxidant capacity were procured from Sigma-Aldrich, Santiago, Chile at ≥98% purity. For fatty acid identification and quantification, a standard fatty acid methyl ester (FAME) mix, C4–C24, by Supelco Analytical (Bellefonte, PA, USA) was used, and tripentadecanoin >99% (Nu-Check Pre, Inc., Elysian, MN, USA) was used as the internal standard. 

### 3.2. Total Carotenoid Extraction and Quantification

Carotenoids were extracted by using a slightly modified version from Montero, Calvo, Gómez-Guillén and Gómez-Estaca [32]. Briefly, 5 mL pure *n*-hexane was used to extract per 10 mg oleoresin samples under dark conditions. The extracts were mixed with more solvent, shaken vigorously, and centrifuged (3290× *g*, 5 min) until the oleoresin biomass was colorless. Carotenoid content was determined by the spectrophotometric method (Shimadzu UV-1280, Kyoto, Japan) in triplicate according to the method used by Britton [77]. The carotenoid extracts were also analyzed by HPLC after evaporating the previous solvent and re-suspending it in acetone.

### 3.3. Sample Preparation and Enzymatic Hydrolysis

The samples for hydrolysis were prepared as described by Mezquita et al. [78,79], and, for enzymatic hydrolysis, the method reported by Lorenz [80] was used with some modifications. For quantifying all the astaxanthin present in the oleoresin (free, mono-, and diesters), it was necessary to break the ester bonds of this carotenoid. In particular, the enzyme cholesterol esterase was used for specific ester bond hydrolysis, providing a highly efficient way to break these bounds. A 4 unit/mL stock of cholesterol esterase solution in 0.05 M Tris-HCl pH 7.0 buffer was prepared and maintained at 5 ±2 °C. Then, 3 mL of the sample was transferred to an empty test tube, to which 3 mL of the enzyme solution was added, and the test tube was then capped in triplicate. The test tubes were placed in a water bath accurately regulated to 37 °C with gentle stirring for 45 min. Thereafter, 1.0 g of anhydrous sodium sulfate was added, astaxanthin was extracted with 2 mL of hexane, and the sample was re-dissolved in 3 mL of acetone and then analyzed by high-performance liquid chromatography (HPLC).

### 3.4. Individual Carotenoid Analysis by High-Performance Liquid Chromatography

The gradient method described by León et al. [81] was used to separate and determine lutein, canthaxanthin, and β-carotene content at 450 nm and astaxanthin content at 471 nm detection wavelengths. Previously, all carotenoid extracts and hydrolysis extracts were filtered (Ø 0.20 µm polytetrafluoroethylene (PTFE) membrane filter), and carotenoid content in both the extracts was determined by an HPLC system (model 7100 equipped with three pumps, an RP-18 column, and a UV-Vis detector; Merck Hitachi Lachrom, Tokyo, Japan). The mobile phases were ethyl-acetate (as solvent A) and acetonitrile and water (9:1 v/v) (as solvent B). The corresponding calibration curves of the standards were used as reference to identify and quantify the individual carotenoids as mg carotenoids/g of oleoresin sample. 

### 3.5. Antioxidant Capacity

#### 3.5.1. Extract of Oleoresin and Radical Scavenging Activity

Oleoresin samples were dissolved in methanol with a stock concentration of 2.0 mg/mL. Partially dissolved extract was exposed to an ultrasonic bath (Biobase, Digital Ultrasonic Cleaner, Jinan, China), filtered (Ø 0.20 µm PTFE membrane filter), and then used for determining antioxidant capacity. All extracts were maintained at a temperature below −20 °C in the dark until analysis. For determining the radical scavenging activity, assays using DPPH (2,2-diphenyl-1-picrylhydrazyl) developed by Mathew and Subramanian [82] and ABTS (2,2′-azino-bis (3-ethylbenzothiazoline-6-sulfonic acid diammonium salt) by Gasca et al. [83] and Bekir et al. [84] were also performed, but these results were included in this study because there was interference with the color of the oleoresin samples. 

#### 3.5.2. Ferric Reducing Antioxidant Potential Assay

The ferric reducing antioxidant potential (FRAP) assay was performed as described by Akter et al. [85]. The FRAP reagent was prepared using 300 mM acetate buffer (pH 3.6, 10 mM of 2,4,6-tris-(2-pyridyl)-s-triazine (TPTZ) solution in 40 mM hydrochloric acid) and 20 mM FeCl_3_·6H_2_O aqueous solution in the ratio of 10:1:1 (v/v). The extracts were prepared at a final concentration of 200 μg/mL. Extract solution (10 μL) was mixed with 70 μL of freshly prepared FRAP reagent and incubated at 37 °C for 30 min. The absorbance of the solutions was measured at 593 nm using a microplate reader (BioTek Synergy HTX multi-mode reader, software Gen5 2.0, Winooski, VT, USA). Trolox was used as the standard to prepare the calibration curve in a concentration range of 0–1.0 mg/mL. The results of FRAP assay were expressed in mg of Trolox equivalents (TE)/g of oleoresin extract (OE). 

#### 3.5.3. Oxygen Radical Absorbance Capacity Assay

The oxygen radical absorbance capacity (ORAC) assay was performed according to the method reported by Wu et al. [86] and developed by INTA (Institute of Nutrition and Food Technology, University of Chile, Chile). The results of the assay were expressed in µmol TE/100 g of OE.

### 3.6. Determination of Total Phenol Content

The 96-well microplate Folin–Ciocalteu method described by Ainsworth and Gillespie [87] was used to determine the total phenol content. A total of 20 μL of the diluted extract (2.0 mg/mL) was mixed with 100 μL of 10% (v/v) Folin–Ciocalteu reagent and stirred. The mixture was left for 5 min, and then 75 μL of sodium carbonate solution (700 mM) was added, and the mixture was stirred for 1 min. After 60 min, absorbance was measured at 765 nm and at room temperature by using a microplate reader (BioTek Synergy HTX multi-mode reader, software Gen5 2.0, Winooski, USA). For ensuring the accuracy of the absorbance values of the extracts, we determined the absorbance value of the same reaction using methanol (blank or control), instead of the extract or standard, and subtracted this value from the absorbance value of the reaction with the sample. Diluted gallic acid (0–1000 μg/mL) was used as the standard for calibration. The results were expressed in mg of gallic acid equivalent (GAE)/g of OE.

### 3.7. Fatty Acid Extraction

The extraction of fatty acid methyl esters (FAMEs) was performed according to a modified version of the direct acid catalysis method described by Lamers et al. [88]. The reaction mixture containing 10 mg of oleoresin biomass, 10 ppm of internal standard, and 3 mL solution of 5% (v/v) H_2_SO_4_ in methanol was incubated at 80 °C for 1 h with continuous agitation. Then, the flasks were washed with hexane and Milli-Q water until the pH of the water after washing was neutral. The mixture was separated into two layers by centrifugation (360× *g*, 10 min). The upper oil layer (FAMEs diluted in hexane) was separated and washed with Milli-Q water for further analysis and quantification by gas chromatography. 

### 3.8. Fatty Acid Analysis by Gas Chromatography

A gas chromatograph (Shimadzu 2010, Kyoto, Japan) equipped with a flame ionization detector (FID) and a split/splitless injector was used to analyze fatty acid composition. In all cases, samples (1 μL) were injected into a capillary column (RESTEK; 30 m, 0.32 mm i.d., 0.25 μm film thickness). The injector temperature was maintained at 250 °C in the split mode with a split ratio of 4.5:1, and nitrogen was used as the carrier gas at a constant flow rate of 11.25 mL·min^−1^. The oven temperature was maintained at 80 °C for 5 min, increased to 165 °C at 4 °C·min^−1^ for 2 min, and further increased to 180 °C at 2 °C·min^−1^ for 5 min. It was heated at a rate of 2 °C·min^−1^ to 200 °C for 2 min. It was re-heated at a rate of 4 °C·min^−1^ to 230 °C for 2 min and finally maintained at that temperature for 2 min, reaching 250 °C at 2 °C·min^−1^. The detector temperature was 280 °C. Individual FAMEs were identified by comparing their retention times with those of mixed FAME standards (FAME Mix C4-C24, Supelco Analytical) and quantified by comparing their peak area with that of mixed FAME standards as well as an internal standard (tripentadecanoin ~10 ppm/sample, Nu-Check Pre, Inc., Elysian, MN, USA).

### 3.9. Viscosity Measurements

Oleoresin samples were weighed into a metallic cylindrical flask to control viscosity. A rotational type Brookfield viscometer (model RVDV-II+, Brookfield Engineering Laboratories, Stoughton, MA, USA) was used at a rotational speed that ranged between 20 and 100 rpm. All assays were performed with the small sample adapter and the spindles SC-21 and SC-27, and the sample volumes were 7.1 and 10.5 mL for each spindle, respectively. The assays were performed at different temperatures (a range of 5–20 ± 0.1 °C in steps of 5 °C, called the “low-temperature zone”, and a range of 30–60 ± 0.1 °C in steps of 10 °C, denoted as the “high-temperature zone”), controlled by a shaking water bath (model YCW-012, Gemmy Industrial Corporation, Taipei, Taiwan). Viscosity behavior was derived mathematically with the help of the power law model, $\tau =K{\dot{\gamma }}^{n}$, where τ (Pa) is the shear stress, *K* (Pa s^n^) the consistency index, $\dot{\gamma }$ (s^−1^) the shear rate, and n (dimensionless) is the behavior index of the fluid. To control the effect of temperature on viscosity, apparent viscosity was calculated based on the Arrhenius equation, $\eta_{a}=\;\eta_{0}*e^{\frac{E_{a}}{RT}}$, where $\eta_{a}$ (Pa s) is the apparent viscosity of the oleoresin samples, $\eta_{0}$ (Pa s) is a frequency factor, $E_{a}\;$(kJ/mol) is activation energy, R (8.314 J/mol K) is the universal molar gas constant, and T (°K) is the absolute temperature. 

### 3.10. Statistical Analysis

The presented values, unless otherwise indicated, were means of three independent measurements from several oleoresin samples. The standard deviations of each set of experiments are represented in the corresponding Figure 1, Figure 2 and Figure 3 and Table 1, Table 2, Table 3, Table 4 and Table 5 (*p < 0.05*; *n* = 3). Statistical evaluation of the results was performed using common statistical methods by using the IBM SPSS Software. For viscosity analysis, the equations described were determined with the help of STATGRAPHICS Centurion XVI (Version 16.1.03) software (StatPoint Technologies Inc., Warrenton, VA, USA). The determination coefficient (R^2^) was one of the main criteria to evaluate the fit quality of proposed models, complemented by calculating the root mean square error (RMSE), sum square errors (SSE), and reduced chi-squared (χ^2^) statistical parameters.

## 4. Conclusions

This study indicates the potential of *H. pluvialis* oleoresin as a source of commercially valuable compounds and/or functional ingredients for today’s food industry. We found that astaxanthin was present in the highest concentration in all its forms (free astaxanthin and astaxanthin esters), followed by β-carotene. These results indicate that food products derived from microalgae through green technology-based extraction could contribute as a viable option of nutrient-rich food products. Total phenolic content, fatty acid profile, and antioxidant capacity of our oleoresin samples also confirmed that this novel product could open a range of possibilities to be applied in the health food industry. Moreover, through our viscosity studies, we have been able to further characterize the oleoresin produced from *H. pluvialis*, while the Arrhenius model helped us understand and document the effect of temperature on the apparent viscosity of oleoresin. Therefore, we conclude that these results can be considered relevant to food, pharmaceutical, and cosmetic industries because of the important properties of the *H. pluvialis*-based oleoresin to these industries and the scarcity of related studies reported the literature. Thus, we believe that the antioxidant and viscosity properties of a carotenoid-rich oleoresin extracted by the SC-CO_2_ method from *H. pluvialis* could be utilized in a variety of ways in these industries, especially the food industry.

## Figures and Tables

**Figure 1 molecules-24-04073-f001:**
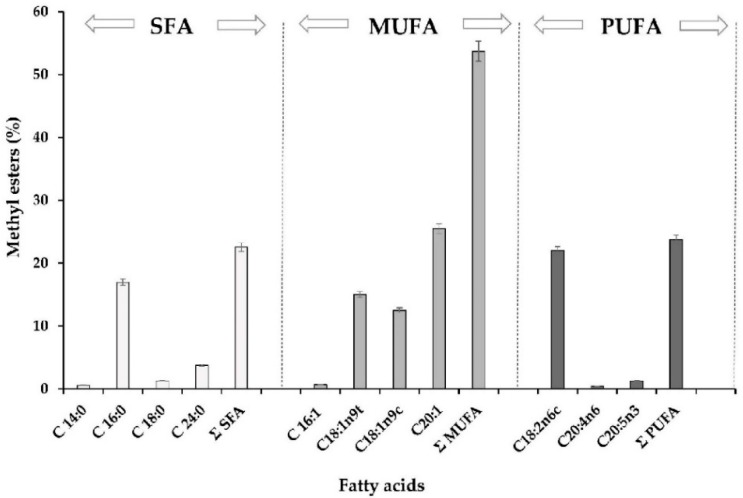
Fatty acid composition of the oleoresin extracted by SC-CO_2_ from the microalga *Haematococcus pluvialis* determined by GC-FID. SFA: saturated fatty acids, MFA: monounsaturated fatty acids, and PFA: polyunsaturated fatty acids. All data are measured as the percentage of total fatty acid methyl esters (% area; *n* = 3; ±SD).

**Figure 2 molecules-24-04073-f002:**
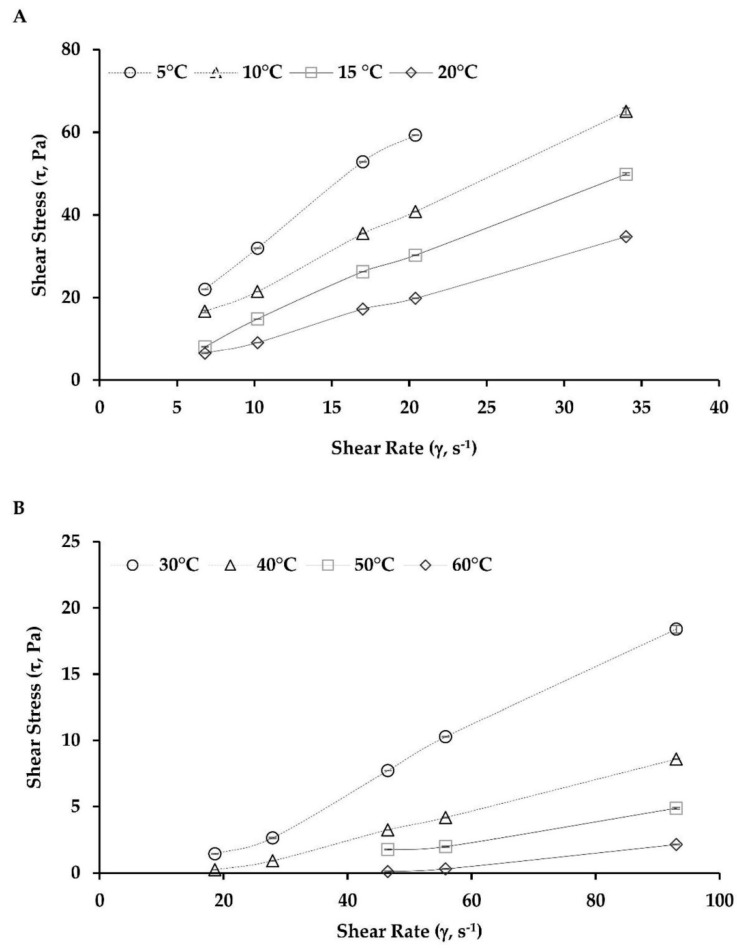
Rheogram flow curve of oleoresin at different temperatures with a relationship between shear stress (τ; Pa) and shear rate (γ; s^−1^). (**A**) Low temperatures from 5 to 20 °C (with spindle SC–21; *n* = 3) and (**B**) high temperatures from 30 to 60 °C (with spindle SC–27; *n* = 3).

**Figure 3 molecules-24-04073-f003:**
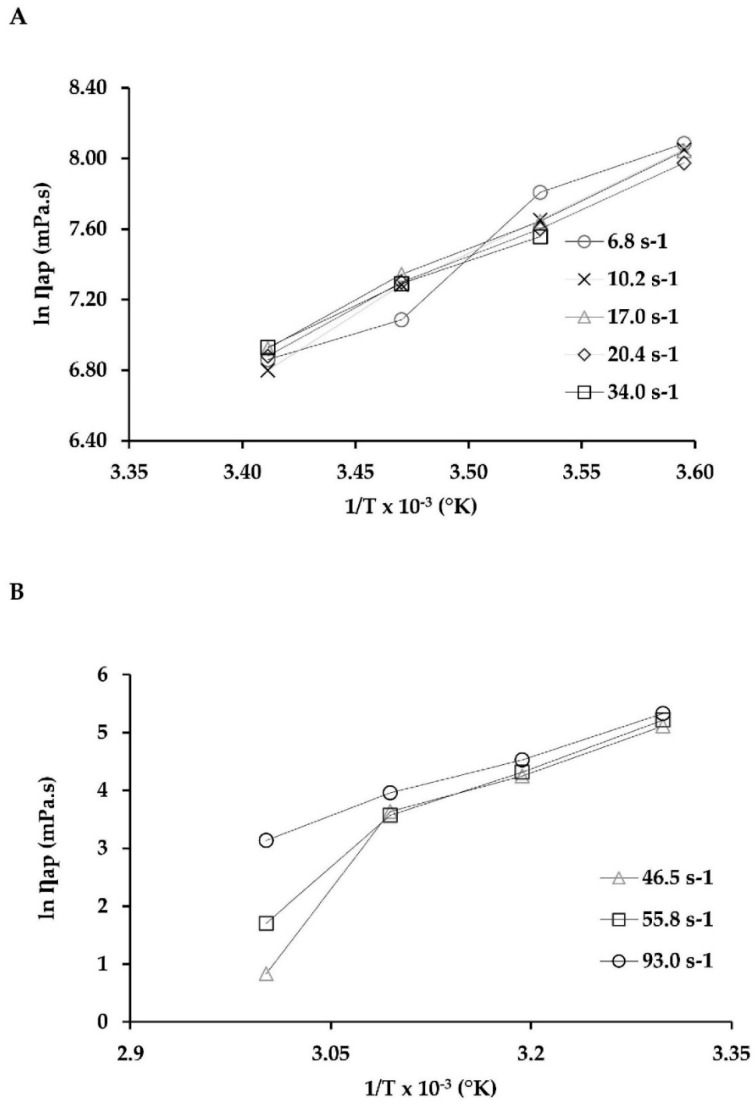
Temperature effect on the apparent viscosity of oleoresin at different shear rates. (**A**) Low temperatures from 5 to 20 °C (with spindle SC–21; *n* = 3) and (**B**) high temperatures from 30 to 60 °C (with spindle SC–27; *n* = 3).

**Table 1 molecules-24-04073-t001:** Quantification of individual and total carotenoids in oleoresin extracted by the SC-CO_2_ method from the microalga *Haematococcus pluvialis* using analytical method and HPLC. The results were expressed as mg of carotenoids/g oleoresin and as the relative abundance of total carotenoid (%wt) (± SD, *n* = 3).

Carotenoid	Specific Name	Content in Oleoresin (mg/g)	Abundance (%wt)
Free astaxanthin	3,3′-dihydroxy-β, β-carotene-4,4′-dione	6.82 ± 0.19	5.94
Canthaxanthin	β, β-carotene-4,4′-dione	1.68 ± 0.19	1.46
β-carotene	β, β-carotene	2.35 ± 0.29	2.05
Lutein	β, ε-carotene-3,3′-diol	1.12 ± 0.23	0.97
Total astaxanthin	Free-astaxanthin and astaxanthin esters *	96.22 ± 1.21	83.76
Others	-	6.68 ± 0.30	5.81
**Total carotenoids**		**114.9 ± 0.89**	**100.0**

* Astaxanthin esters as mono- and diesters by the cholesterol esterase hydrolysis method in *H. pluvialis* oleoresin.

**Table 2 molecules-24-04073-t002:** Quantification of antioxidant capacity and total phenols in oleoresin extracted by SC-CO_2_ from the microalga *Haematococcus pluvialis* using various analytical methods.

Analytical Determination		Unit	Value	±SD
**Antioxidant capacity**	**FRAP**	mg TE/g OE	313.76	5.92
	**ORAC**	µmol TE/100 g OE	5.22	0.16
**Total phenols**	-	mg GAE/g OE	74.08	3.29

TE, Trolox equivalents; OE, oleoresin extract; GAE, gallic acid equivalent; FRAP, ferric reducing antioxidant potential; ORAC, oxygen radical absorbance capacity.

**Table 3 molecules-24-04073-t003:** Rheological parameters by the power law model for oleoresin extracted at different temperatures by SC-CO_2_ from the microalga *Haematococcus pluvialis*.

Temperature		*η* (Pa s)	*K* (mPa s^n^)	*n*	R^2^	χ^2^	SSE	RMSE
Zones	(°C)							
Low	5	3.10 ± 0.12	3800.45	0.92	0.998	3.41 × 10^−03^	4.55 × 10^−03^	0.067
	10	2.11 ± 0.18	3085.98	0.85	0.995	5.33 × 10^−03^	6.66 × 10^−03^	0.082
	15	1.43 ± 0.12	1043.79	1.11	0.989	8.08 × 10^−03^	1.01 × 10^−02^	0.101
	20	0.97 ± 0.05	824.25	1.06	0.997	1.01 × 10^−03^	1.27 × 10^−03^	0.036
High	30	0.15 ± 0.05	12.12	1.64	0.986	2.58 × 10^−04^	3.23 × 10^−04^	0.018
	40	0.06 ± 0.03	0.47	2.23	0.965	2.78 × 10^−04^	3.47 × 10^−04^	0.019
	50	0.04 ± 0.01	4.53	1.54	0.977	4.04 × 10^−04^	6.06 × 10^−04^	0.025
	60	0.01 ± 0.01	1.36 × 10^−05^	4.17	0.990	1.37 × 10^−06^	2.06 × 10^−06^	0.001

All data are expressed as η = apparent viscosity, K = consistency index, and n = flow behavior index.

**Table 4 molecules-24-04073-t004:** Parameters of the Arrhenius equation for the oleoresin extracted by SC-CO_2_ from the microalga *Haematococcus pluvialis* at low temperatures (5–20 °C).

Shear Rate (s^−1^)	$\eta_{0}\;(\mathrm{Pa}\;s)$	$E_{a}\;(\mathrm{kJ}/\mathrm{mol})$	R^2^
6.8	4.82 × 10^−08^	57.70	0.946
10.2	2.04 × 10^−07^	54.24	0.957
17.0	2.65 × 10^−06^	48.26	0.947
20.4	4.22 × 10^−06^	47.03	0.974
34.0	1.74 × 10^−05^	43.65	0.984

All data are expressed as frequency factor ($\eta_{0}$) and activation energy ($E_{a}$).

**Table 5 molecules-24-04073-t005:** Parameters of the Arrhenius equation of oleoresin extracted by SC-CO_2_ from the microalga *Haematococcus pluvialis* at high temperatures (30–60 °C).

Shear Rate (s^−1^)	$\eta_{0}\;(\mathrm{Pa}\;s)$	$E_{a}\;(\mathrm{kJ}/\mathrm{mol})$	R^2^
18.6	-	-	-
27.9	-	-	-
46.5	2.61 × 10^−17^	110.08	0.863
55.8	2.37 × 10^−14^	92.71	0.934
93.0	1.18 × 10^−8^	59.47	0.992

All data are expressed as frequency factor ($\eta_{0}$) and activation energy ($E_{a}$), and “-" indicates torque values < 10%.

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
