# Peer review of "Determining the Potential of Haematococcus pluvialis Oleoresin as a Rich Source of Antioxidants"

_molecules, 2019, doi:10.3390/molecules24224073_

Round 1
Reviewer 1 Report
The manuscript presented by Ruiz-Dominguez shows a very good analysis of a H. pluvialis extract.
I really miss the extraction conditions of the extract and the composition, at least proximal, of other components in the oleorresin. In fact, the composition of the oil used to prepare the oleoresin will determine the reuslts obtained in section 2.3. So it's very important.
The rest of the manuscript is quite good. It's a good point to compare your oleoresin with other resins like pine. Please if you find some data using a more similar would be better (somekind of marine oleoresin).
Regarding to the carotenoid composition, let me recommend for future works to use a C30 HPLC column, the resolution of carotenoids is spectacular. Taking into acount that the oleoresin is 10% extract, the values showed in table 1 are very good, even better than other works dealing with Supercritical fluid extraction of H. pluvialis such as Reyes et al. 2014 https://doi.org/10.1016/j.supflu.2014.05.013 in this sense a good point for your work could be to include a brief comparison of the astaxanthin in your oleoresin and other extracts found in literature.
And finally, please clarify why you used the enzymatic hydrolysis described in 3.3. I supose it's to break the ester bound astaxanthin-fatty acid but it's not written.
Good work!
Author Response
Please see the attachment denominated "Respond 1".
Thanks

Reviewer 2 Report
Nowadays, consumers are more health conscious about what they eat and functional food is becoming more and more relevant. Nutraceuticals have physiological benefits and some of them could be extracted from marine organisms such as microalgae. Microalgae have been used over the years for human consumption in various forms such as tablets, capsules, and liquids. Haematococcus pluvialis is a microalgae that synthesizes and accumulates high level of astaxanthin in nature. This microalgae has been widely used due to its antioxidant and colorant properties.
Authors of presented manuscript produced the H. pluvialis oleoresin and studied it as an potential alternative for developing functional ingredients for designing healthy food products. However, the subject is not new – there are Astaxanthin-Rich Oleoresins from Haematococcus pluvialis which are authorized to use in China as well as EU market (such as Zanthin®). Authors must discussed how they experiments differ from the already published data (before and during the process of approval of presented supplements). Also, the novelty of the work should be highlighted (the precise area of the microalgae collection is not enough for the novelty statement).
The work is well organized, the analytical procedures are chosen correctly. The authors should place the explanation of the shortcuts used as they first appear in the text.
There are some spelling mistakes (Paragraph 2. 2.1. line 104 SS-CO) and the text should be studied more carefully for those mistakes.
Author Response
Please see the attachment denominated "Respond 2fv"
Thanks

Round 2
Reviewer 2 Report
Thank you for reviewing carefully my comments. After those changes J recommend the manuscript to be published in Molecules.